# How to Estimate the Probability of Tolerance Long-Term in Liver Transplant Recipients

**DOI:** 10.3390/jcm12206546

**Published:** 2023-10-16

**Authors:** Dennis Eurich, Stephan Schlickeiser, Ramin Raul Ossami Saidy, Deniz Uluk, Florian Rossner, Maximilian Postel, Wenzel Schoening, Robert Oellinger, Georg Lurje, Johann Pratschke, Petra Reinke, Natalie Gruen

**Affiliations:** 1Department of Surgery, Charité—Universitätsmedizin Berlin, 13353 Berlin, Germany; ramin-raul.ossami-saidy@charite.de (R.R.O.S.); deniz.uluk@charite.de (D.U.); maximilian.postel@charite.de (M.P.); wenzel.schoening@charite.de (W.S.); robert.oellinger@charite.de (R.O.); georg.lurje@charite.de (G.L.); johann.pratschke@charite.de (J.P.); 2Berlin Institute of Health (BIH) Center for Regenerative Therapies (BCRT), Charité—Universitätsmedizin Berlin, 13353 Berlin, Germany; stephan.schlickeiser@bih-charite.de (S.S.); petra.reinke@charite.de (P.R.); 3Department of Pathology, Charité—Universitätsmedizin Berlin, 13353 Berlin, Germany; florian.rossner@charite.de; 4Max Delbrueck Center for Molecular Medicine, Helmholtz Association, 13125 Berlin, Germany; 5Berlin Center for Advanced Therapies (BeCAT), Berlin Institute of Health (BIH) Center for Regenerative Therapies (BCRT), Charité—Universitaetsmedizin Berlin, 13353 Berlin, Germany; natalie.gruen@charite.de; 6Department of Nephrology and Internal Intensive Care Medicine, Charité—Universitätsmedizin Berlin, 13353 Berlin, Germany

**Keywords:** liver transplantation, operational tolerance, graft loss, immunosuppression, graft rejection

## Abstract

Background: Operational tolerance as the ability to accept the liver transplant without pharmacological immunosuppression is a common phenomenon in the long-term course. However, it is currently underutilized due to a lack of simple diagnostic support and fear of rejection despite its recognized benefits. In the present work, we present a simple score based on clinical parameters to estimate the probability of tolerance. Patients and methods: In order to estimate the probability of tolerance, clinical parameters from 82 patients after LT who underwent weaning from the IS for various reasons at our transplant center were extracted from a prospectively organized database and analyzed retrospectively. Univariate testing as well as multivariable logistic regression analysis were performed to assess the association of clinical variables with tolerance in the real-world setting. Results: The most important factors associated with tolerance after multivariable logistic regression were IS monotherapy, male sex, history of hepatocellular carcinoma pretransplant, time since LT, and lack of rejection. These five predictors were retained in an approximate model that could be presented as a simple scoring system to estimate the clinical probability of tolerance or IS dispensability with good predictive performance (AUC = 0.89). Conclusion: In parallel with the existence of a tremendous need for further research on tolerance mechanisms, the presented score, after validation in a larger collective preferably in a multicenter setting, could be easily and safely applied in the real world and already now address all three levels of prevention in LT patients over the long-term course.

## 1. Introduction

To avoid rejection after liver transplantation (LT), individual immunosuppression (IS) is necessary. Potent IS leads to a low graft loss rate (10–15%) on the one hand, but on the other hand it leads to a significantly increased probability of experiencing the reverse side of the coin such as infections, cardiovascular events, tumor diseases, kidney diseases, and diabetes complications, which is why setting an optimal IS level is a balancing act [1,2].

Compared to other transplantable organs, the liver is privileged due to numerous immunologic properties, and therefore requires much lower IS levels, and in the long-term, up to no IS at all, without functional impairment [3]. Operational tolerance corresponds to the ideal state of unrestricted organ function in the absence of IS side effects [3,4].

The long-term impact of acute rejection (ACR) on graft and patient survival is controversial. While in the postoperative period the concern for rejection is dominant and a balance to infection is sought, the long-term transplant patient is in a stable state in which acute rejection is rare [5,6]. In carefully selected patients without autoimmune-mediated underlying liver disease, the rate of spontaneous tolerance may be as high as 79% in the long-term [7].

According to the European Liver Transplant Registry, approximately 160,000 LTs have been performed since its inception. Half of these are still alive and struggling with the consequences of the long-term toxicity of IS [1]. The risk of developing malignant tumors depends on the cumulative duration and intensity of IS used and is increased by a factor of 2.6 to 4.3 compared with the normal population [8,9,10]. Approximately 30% of all LT patients will die from either tumor recurrence, mostly hepatocellular carcinoma (HCC) or de novo tumor [1]. Recently, our group has shown that the restrictive use of IS in patients with de novo lung cancer and recurrent HCC may significantly prolong survival and may well be conceived as a useful measure in an already dramatic situation [11,12]. Furthermore, LT patients without IS respond significantly better to standard SARS-CoV-2 vaccination [13]. If spontaneous tolerance can be identified early, advantages are obvious. 

Studies conducted to date on controlled IS discontinuation have yielded partly contradictory results mostly due to non-uniform definitions of tolerance. In the absence of histological confirmation, which is still considered the gold standard in the diagnosis of liver disease, a considerable limitation arises in validation trials. Analyses of complex expression patterns of various biomarkers and subcellular features that are laborious to determine prove to be uncertain and, above all, impractical to handle in order to predict tolerance and rejection [14,15,16,17]. Therefore, reliable and preferably non-invasive biomarkers are urgently needed [18]. Time since LT, age at the moment of LT, and male gender seem to be associated with tolerance [7,19]. 

The aim of this work is to develop a score consisting of clinical variables for the easy estimation of tolerance in LT patients in the long-term course. 

## 2. Patients and Methods

Demographics: More than 3000 LTs have been performed at our institution (Charité Universitaetsmedizin Berlin) since 1988. All patients are routinely followed-up at our outpatient center according to the institutional protocol ranging from twice a week to once in three months depending upon the time passed since LT, lifelong. 

Protocol biopsies were performed beginning with one year after LT alternating every 2 to 3 years for indefinite time or upon indication, in order to adjust the IS extent in an individual manner by adhering to the motto “as little as possible, as much as necessary”. Calcineurin inhibitors (CNI) and the steroid tapering regimen were the backbone of the IS with or without antiproliferatives and the mammalian target of rapamycin inhibitors (mTORi) adapted to the individual comorbidity and tolerability. The use of induction by thymoglobulin or basiliximab was reserved for cases of retransplantation and for patients with autoimmune-mediated underlying disease. In the case of oncological diseases such as HCC, we do not use induction.

The indication for LT corresponded to the basic etiological diagnoses that led to the end stage liver disease (ESLD), which were subdivided as alcoholic, hepatitis C (HCV) and hepatitis B (HBV) virus associated, cholestatic liver diseases, and others. Furthermore, a classification was made according to the abruptness of liver failure and classified as follows: acute liver failure, chronic liver failure in the sense of cirrhosis, and HCC in cirrhosis.

According to the clinical, laboratory, and histological assessment of the course, weaning was performed clinically on an ongoing basis. Patients were identified who (i) weaned or discontinued IS independently, (ii) weaned under controlled conditions, and (iii) had IS discontinued as ultima ratio due to malignancy. IS was reduced gradually in several steps until either discontinuation was possible or rejection occurred.

The strictest definition of tolerance was clinical, laboratory, histological stability of the graft function and patient at least one year after discontinuation of IS. If histological examination was not possible due to the anticoagulation used, the fibrosis stage was determined by Fibroscan before and after at the earliest one year off the IS. The elevation of transaminases was not allowed to be higher than double the individual baseline. 

If clinical rejection was suspected, histologic confirmation was performed and, if confirmed according to Banff-classification, a pulse steroid therapy was given and the maintenance dose of IS was resumed [20].

Desmet and Scheuer classification was selected for the grading of inflammation and staging of fibrosis because of its superiority in reproducibility over other semiquantitative systems of fibrosis assessment [21,22]. Fibrosis staging was performed using a scale of 0–4 (0: none, 1: minimal without septa, 2: moderate with few septa, 3: portoportal septa without cirrhosis, and 4: cirrhosis). Grades of inflammation were classified as 0: no inflammation; 1: minimal; 2: mild; 3: moderate; and 4: severe. The content of fat was classified as no fat if less or equal to 5% in the histology, 5–33% as stage 1, between 33 and 66% as stage 2, and more than 66% as stage 3 [23].

## 3. Statistical Analysis

Clinical, laboratory, and histological data were collected from a prospectively organized database and patient records for the analysis with the commercially available statistical software SPSS version 27 (Chicago, IL, USA). Differences in the distribution of categorical data were tested using Fishers’ exact test. Nonparametric Mann–Whitney U or Kruskal–Wallis tests were applied to detect differences in continuous data when appropriate. The Wilcoxon-test was performed to detect differences in pair-matched parameters before weaning and thereafter. 

All multivariable analyses were conducted in R (version 4.1.1.) using the Hmisc (version 4.6-0) and rms packages (version 6.2-0). For developing a simple predictive scoring model, a logistic regression model was fitted to estimate the probability of tolerance. The full model was prespecified to include the following clinical parameters: time since LT (dichotomized to less than vs. at least 10 years after LT), recipient gender, donor gender, HCC in the explant, post-LT malignancy, monotherapy (CNI, mycophenolate mophetil (MMF) or everolimus (EVL) vs. any combined therapy), history of acute cellular rejection (ACR), and viral cirrhosis (HCV- or HBV-associated ESLD) with model coefficients re-estimated by L2-penalized regression. To simplify the model, linear predictors of the full penalized model were approximated by ordinary least squares regression and factors were removed by fast backward step-down selection using Akaike’s information criterion as a stopping rule. The approximate model retained five variables: time since LT, recipient gender, HCC, monotherapy, and ACR with only a minute loss of precision (R^2^ = 0.96 against full model). Correct estimates of the variance of the approximate coefficients were calculated using equation 5.2 [24].

A final simple points score could be depicted as a nomogram with coefficients rounded to integers for each clinical characteristic (precision R^2^ = 0.95 against full model). Receiver operator characteristics were analyzed using the *pROC* package (version 1.18.0). Performance measures were calculated for a threshold that optimizes sensitivity and specificity (“closest top-left” method) and corresponding 95% confidence intervals (CI) were computed with 2000 stratified bootstrap replicates.

The study was performed retrospectively according to the professional code of conduct of the German Medical Association (Article B.III.§15) on the basis of the Declaration of Helsinki of the World Medical Association and approved by the ethics committee of Charité Universitaetsmedizin Berlin (protocol code EA1/035/21).

## 4. Results 

Demographics: Among 2985 patients, a group of 82 individuals with an immunologically neutral baseline diagnosis met the criteria for inclusion in the analysis (Table 1). Fifteen (18.3%) patients discontinued IS on their own without timely consulting the transplant center. Reasons were noncompliance, excessive self-initiative, and fear of IS side effects. Forty-eight (58.5%) patients underwent weaning in a closely controlled mode whereas the discontinuation of IS was completed in a controlled manner as a supportive measure in 19 (23.2%) patients when malignancies occurred. In the overall cohort, IS discontinuation was possible in 58 (70.7%) patients while 24 (29.3%) patients demonstrated graft dysfunction in terms of rejection during the weaning period and had to be assigned to more intensive IS therapy again. Ten patients developed rejection during the first year after IS cessation after 2.5 months in median (1.0–7.0). IS was reintroduced in 8 patients, whereas it was not possible in 2 patients due to profound incompliance (Figure 1).

The driving force to discontinue IS was side effects and or fear of them, which, among other factors, best mapped noncompliance with the prescribed medication. Table 2 lists the most common side effects of common immunosuppressants that occurred in the cohort. On average, the number of IS-associated adverse events was 2 per person, with only a minority of 10 (12.2%) patients developing no adverse events.

A total of 48 (58.5%) patients were tolerant according to the clinical dispensability of IS without evidence of graft dysfunction either until death in the face of malignancy or at least until reaching the first IS-free year. The median duration of observation of tolerant patients since the discontinuation of immunosuppression was 45.4 (17.3–255.2) months following the definition of the minimum duration of 12 months including all deceased patients. Among 15 patients who admitted unauthorized IS discontinuation, the rate of tolerance was as high as 73.3% based on 11 patients who were fine and did not demonstrate any significant signs of graft dysfunction according to the definition. Among 48 patients undergoing IS weaning disciplined in a controlled manner, 23 (47.9%) patients demonstrated tolerance while 25 (52.1%) patients developed acute cellular rejection requiring IS reinstitution. However, the rate of tolerant patients with tumor development (total 14/19 patients) demonstrated a remarkable tolerance rate of 73.7%.

### 4.1. Levels of Tolerance Definition Accuracy

Fibrosis stages were assessed in 81 (98.8%) patients before weaning and in 65 (79.3%) at least one year after, according to the definition of tolerance. The diagnosis of tolerance or intolerance was based on 62 (75.6%) histologically, in 4 (4.9%) patients using Fibroscan because rejection was not even suspected, and in 16 (19.5%) patients because of biopsy refusal, anticoagulation, and further logistic difficulties during the COVID-19 pandemic. There was no significant dynamic in fibrosis stages before the initiation of weaning and thereafter according to the Wilcoxon test (*p* = 0.414). A total of 41 patients kept their initial fibrosis stage, 14 patients demonstrated a higher stage, and 10 patients a lower fibrosis stage after weaning. Unsurprisingly, a significant increase in inflammation grade was observed because of acute rejection provoked in non-tolerant patients. Twenty-four patients kept the inflammation grade of the weaning initiation, 31 patients demonstrated a higher inflammation grade, and only 6 a lower inflammation grade compared to the initial histology (*p* < 0.001). No changes in inflammation grade were observed in tolerant patients according to the definition (*p* = 0.593). Interestingly, there was a significant decrease in fat content in the attempt of lowering and weaning of the IS (*p* = 0.012) as displayed in Table 3.

### 4.2. Parameters Associated with Tolerance

The actual age at the time of weaning was the sum of the age at the time of LT and the time since LT in years, and differed significantly among tolerant and nontolerant patients (68.0 vs. 60.9 years; *p* = 0.020). Furthermore, the time since transplantation was significantly longer in tolerant patients than in non-tolerant patients (13.8 vs. 9.2 years; *p* = 0.005). The results are displayed in Table 4.

The proportion of tolerant patients was significantly higher among males than females ((36/49) 73.5% vs. (12/33) 36.4%; *p* = 0.001). Male gender of the donor was also found more frequently than female gender though insignificantly in tolerant patients ((33/50) 66.0% vs. (15/32) 46.9%; *p* = 0.110). The highest rate of rejectors was found among the female recipients of female donor organs ((13/18) 72.2%) compared to female recipients of male organs ((8/15) 53.3%), and the lowest rates were in male patients of female donor livers ((4/14) 28.6%) and male donor liver ((9/35) 25.7%). 

There were 24 (29.3%) patients with HCC in the explant pathology, and 1 patient had been successfully treated for seminoma before LT. The presence of an HCC in explant pathology was significantly associated with tolerance (39.6% vs. 17.4%; *p* = 0.025).

A total of 29 (35.4%) out of all patients developed any kind of malignant disease including 8 (27.6%) cases with cutaneous lesions, 6 (20.7%) patients with posttransplant lymphomas, and 15 (55.2%) patients with solid tumors (Table 2). Among 15 patients with de novo solid tumors, lung cancer was the most frequent (n = 6; 40.0%), followed by malignant tumors of the lower gastrointestinal tract (n = 5; 30.0%). De novo tumor diseases were significantly associated with the tolerant state ((23/29) 79.3% vs. 47.2%; *p* = 0.005).

An absence of documented rejection events since LT were more common in tolerant than in intolerant patients and were significantly associated with tolerance ((33/44) 75.0% vs. ((15/39) 39.5%; *p* = 0.002).

The mode of the last IS was significantly differently distributed among tolerant and intolerant patients in favor of any kind of monotherapy (*p* < 0.001). A total of 31/40 (73.2%) patients on monotherapy with either tacrolimus or MMF or EVL were tolerant. The highest proportion of intolerant patients ((18/24) 75.0%) was found in the group treated with tacrolimus and MMF in combination.

Interestingly, de novo malignancy was found to be significantly more frequent ((13/24) 54.2% vs. (16/58) 27.6%; *p* = 0.041) in patients with HCC presence in the explant pathology compared to patients without HCC.

At the time of analysis, 14 patients had died. Five patients developed advanced tumor disease (4 before weaning and 1 after), 5 patients died of graft failure including one patient who developed SARS-CoV-2-associated cholestatic graft dysfunction. Another 2 developed severe pneumonia and the last 2 died as a result of cerebrovascular disease. Seven (50.0%) patients were tolerant according to the definition. 

### 4.3. Multivariable Analysis and Clinical Prediction of Tolerant Patients

Using penalized estimation, a predictive logistic regression model was developed including the variables time since LT, recipient gender, donor gender, HCC in the explant, post-LT malignancy, monotherapy, history of rejection, and viral cirrhosis. Internal validation of the full model using bootstrap demonstrated acceptable predictive discrimination with a bias-corrected AUC estimate of 0.861.

Despite a considerable association between some of the potential predictors (Figure 2A), monotherapy (odds ratio 6.32; *p*-value 0.003), recipient gender (odds ratio 3.22; *p*-value 0.043), and history of rejection (ACR odds ratio 3.21; *p*-value 0.031) could be identified as leading factors contributing to the dispensability of immunosuppressive therapy (Table 5). Odds ratios were also high in tolerant patients who discontinued IS at least 10 years after transplantation or had HCC in the explant. Approximated coefficients of these five parameters were retained in a simplified model (Table 5 and Figure 2B). The predictive discrimination for tolerance was good with AUC: 0.889 (CI: 0.826–0.955). Using a predicted probability of 0.635 as classification threshold, the model has an accuracy of 0.817, a sensitivity of 0.813 (CI: 0.688–0.918), and a specificity of 0.824 (CI: 0.677–0.941), corresponding to a positive predictive value (PPV) of 0.867 (CI: 0.783–0.951) and a negative predictive value (NPV) of 0.757 (CI: 0.650–0.875).

To further improve practicality, the predictive model could be presented as a points score system (or nomogram; Figure 2D) without a substantial loss of precision. Here, allocating one or two points each to the non-modifiable factors such as absence of HCC in explant pathology (1P), male gender of recipient (1P), and factors that change with time after LT such as time since LT (at least 10 years; 1P), absence of rejection (1P), and meticulous monotherapy (2P), a simple scoring system was formed to estimate the clinical probability of tolerance or dispensability of immunosuppressive therapy.

## 5. Discussion 

The ideal situation is a functioning graft without side effects of IS [3]. The emergence of tolerance, defined as at least 12 months of IS freedom with clinical, laboratory, and histologic integrity of the graft, is a common phenomenon. In our retrospective analysis of real data from a large series of LT patients, a simple scoring system consisting of clinical parameters is presented to estimate the probability of tolerance. The score is based on two nonmodifiable factors such as gender of the recipient and the presence of the tumor in the explant, as well as three additional course-dependent factors such as the time interval since LT, rejection episodes, and IS-monotherapy. These results are partially confirmatory and complementary and may therefore be of high interest for transplant follow-up clinicians, helping them to understand the situation of IS dispensability with the potential to prevent the harm of long-term IS toxicity.

Rejection remains a serious risk to graft health [25]. However, the balance of rejection-promoting factors shifts in favor of tolerance-promoting factors over time after LT [26]. Tolerance is time dependent and may be observed in up to 79% in selected patients and is confirmed in the present analysis [7,27]. Thus, tolerance is much lower in the first years after LT but is not impossible and may significantly increase further along the course in adults, children, and after living donor LT [7,19,27,28]. Age correlations to the necessary strength of IS do not exist. However, the association of a higher age, though insignificant, and time since LT with tolerance according to the present results is striking and is in line with the prior publications [29].

Previously published work largely suggests complex analyses of rejection or tolerance markers to clusters of parameters obtained at the subcellular level, which are costly, complicated, and furthermore fail validation [17,25]. 

After validation, our simple score for the estimation of tolerance probability could be a straightforward tool for a clinical routine with the key advantage of practicality and clinical proximity, as several scenarios are presented that are clinically most common. It is the patient with non-compliance who, with careful observation without dogmatization, is proven right in 73.3% regarding the dispensability of IS, while the non-tolerant rest must be recognized in time. 

This analysis considers patients who are not included in prospective weaning studies because of comorbidity but urgently need a solution for managing IS because they suffer from its consequences [18,30]. Tumor patients are of particular importance, who make up about one third of the total, who may benefit maximally from IS discontinuation in terms of the tertiary prevention. Minimization and even discontinuation in LT-patients as an adjunctive measure in case of secondary malignancy or tumor recurrence (HCC) can prolong survival significantly, easily, and without cost [11,12]. 

The significance of ACR regarding tumor recurrence or de novo tumor development must be measured against a diminished risk of rejection, without increased risk of graft function failure and enhanced overall survival. Thus, optimizing cancer therapy and graft function improves patients’ long-term prognosis. The strategy of tailored IS must be assessed against an increased risk of malignancy. Moreover, both the success of oncologic therapy and preserving graft function determine patient survival. In view of these observations, the immunologically privileged liver status should be considered in posttransplant oncology because of good long-term survival. 

Clearly, the concern for graft survival must not supersede the survival of the recipient. Rather, an integrated approach is important with regards to organ function and malignancy. Treatment options in case of malignancy are limited and overall survival is curtailed compared to a non-transplant setting as, e.g., checkpoint inhibitors are contraindicated due to risk rejection by unmasking the antigen. Decreased immunosuppression improves patients’ prognosis and overall survival following HCC recurrence post LT. The reduction in IS to complete withdrawal would be highly beneficial in the clinical management of malignancies [12]. Currently, there is scant data regarding the type and extent of immunosuppressive therapy to prevent HCC recurrence or to improve long term prognosis [31]. 

The lack of integration of weaning into the principles of follow-up in clinical practice is clearly seen in oncological patients, although the negative influence of IS on the occurrence of post-transplant malignancies has been known for a long time [32,33,34].

This may have been a consequence of industry-sponsored IS trials, where minimizing IS was not considered and the principal focus was on reducing the rate of rejection [35]. In the challenging circumstances of LT patients with de novo neoplasm, immunologic privilege of the liver might offer a solution without an increased risk for the patient and graft survival. The data support this option, especially as maintaining the dose and type of immunosuppression is iatrogenic and results in reduced overall survival.

To date, no prospective studies exist to assess the discontinuation of IS in the setting of tumor diagnosis after LT except for a recommendation to switch the base of IS to everolimus and a guideline-based oncologic approach. By discontinuing IS, the oncologic patient is unlikely to lose anything, since it is not the transplant function that limits survival as recently published [11,12]. Furthermore, there are fundamental differences between a patient after recent LT and a patient in ultralong-term follow-up after several years to decades.

In most weaning studies, a near-perfect collective has been screened for tolerance characteristics at inter-differential levels, and these selected patients do not correspond to the real-world scenario and represent only an ideal minority [36,37]. Clearly, the study of tolerance as a complex trait must continue to take place under maximally controlled conditions [4,30]. However, practicality must be maintained in terms of ease of implementation, low cost, and high diagnostic quality. 

The risk of subclinical rejections may become clinically apparent, if at all, at longer time periods after occurrence, as the processes of fibrogenesis are slow [19,30]. This would most likely affect younger patients with a fairly high life expectancy after LT. For the majority of patients, however, this condition is not relevant. More relevant are the results of IS long-term toxicity.

Although these findings reflect the real-world scenario in a large LT outpatient department, they may underscore the current evidence of the prognostic value for several clinical parameters. The score might be biased by the selection of clinically very stable patients, so that selection may have been intuitively guided by the likelihood of later success of IS discontinuation using some clinical parameters as indicators. Validation in an independent cohort is therefore necessary.

In conclusion, the presented score could be helpful to determine the probability of tolerance and thus contribute to individualized treatment in the long-term course, potentially preventing harm caused by IS long-term toxicity.

## Figures and Tables

**Figure 1 jcm-12-06546-f001:**
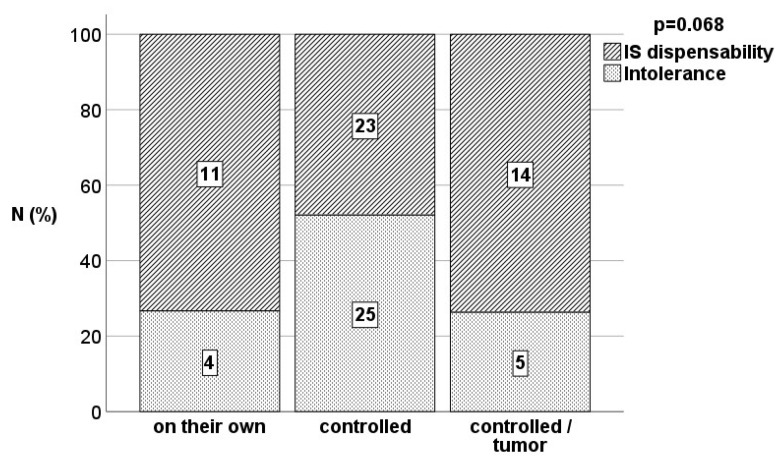
Proportion of tolerant patients and rejectors in the different weaning situations assessed by Fisher’s exact test.

**Figure 2 jcm-12-06546-f002:**
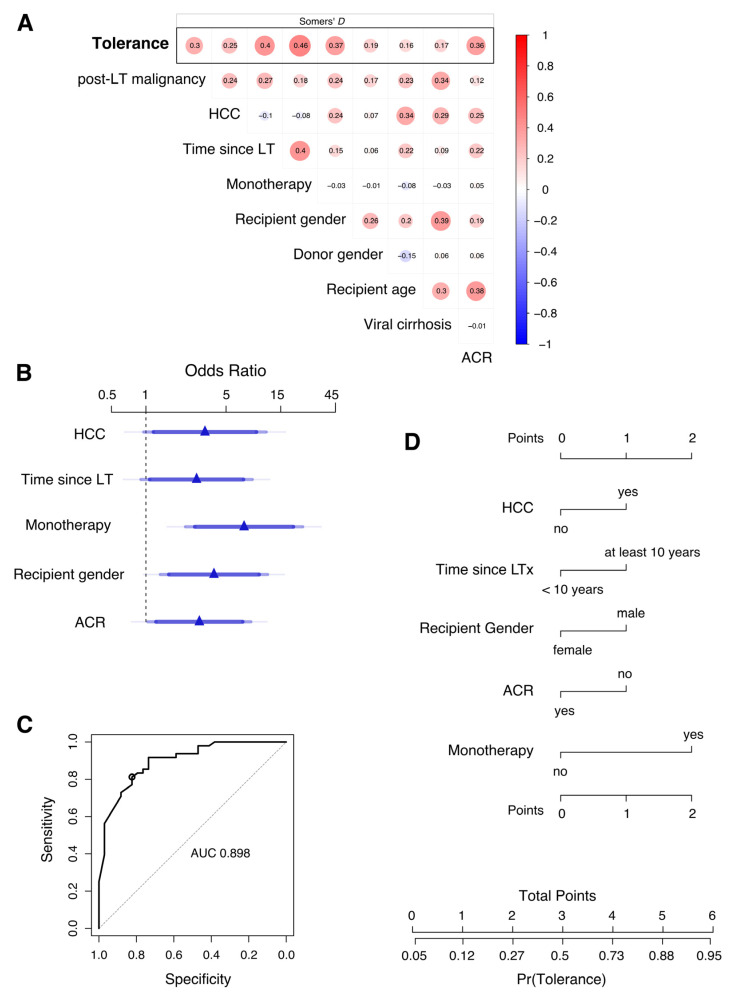
Development of a score to predict the probability of tolerance after discontinuation of immunosuppression in long-term liver transplant patients. (**A**) Pairwise analysis of variables included for multivariable prediction and with tolerance as response (top row). Association between variables is measured by Somers’ D rank correlation. (**B**) Adjusted odds ratios (OR) estimated by penalized multivariable logistic regression analysis are shown for monotherapy, recipient gender, presence of an HCC in the explant, absence of acute cellular rejection, and time since LT. These five predictors were retained in a simplified model that approximates the full model. Effects are shown for the approximate model. Blue of different transparencies indicates 90, 95, and 99% confidence levels. (**C**) Receiver operator characteristic curve for prediction of tolerance using the simplified logistic regression model with approximate coefficients for five variables depicted in (**B**). AUC, area under the curve. The circle indicates sensitivity and specificity for a classification threshold of 0.635 probability. Diagonal segments are produced by ties. (**D**) Nomogram of the approximate predictive model. A points score is calculated based on predictor values to estimate the clinical probability of tolerance.

**Table 1 jcm-12-06546-t001:** Basic demographic characteristics of the cohort (*n* = 82).

Indication for LT; *n* (%)	ALD	13 (15.9)
	HCV	36 (43.9)
	HBV	13 (15.9)
	Cholestatic liver disease	6 (7.3)
	others	14 (17.1)
	Re-LT	10 (12.2)
Mode of liver failure; *n* (%)	acute	5 (6.1)
	cirrhosis	53 (64.6)
	HCC in cirrhosis	24 (29.3)
Last immuno- suppressive medication; *n* (%)	CNI mono	42 (51.2)
	CNI/MMF	23 (28.0)
	MMF mono	13 (15.9)
	CNI/EVL	2 (2.4)
	EVL mono	2 (2.4)
Results of IS discontinuation; *n* (%)	impossible	24 (29.3)
	possible but IS reinstitution	10 (12.2)
	tolerance	48 (58.5)
Mode of weaning; *n* (%)	On their own	15 (18.3)
	Controlled	48 (58.5)
	Controlled (tumor)	19 (23.2)

ALD: alcoholic liver disease, HCV, HBV: ESLD associated with hepatitis C or B virus respectively, HCC: hepatocellular carcinoma, CNI: calcineurin inhibitor, MMF: mycophenolate mofetil, EVL: everolimus.

**Table 2 jcm-12-06546-t002:** Most common immunosuppressants’ side effects in the cohort (*n* = 82); solid tumors include recurrent HCC.

Side Effects of IS	Proportion of Patients; *n* = 82
Arterial hypertension	22 (26.8)
Kidney dysfunction > CKD G3a/b	22 (26.8)
Diabetes mellitus	19 (23.2)
Infections	40 (48.8)
Gastrointestinal disturbances	12 (14.6)
Neurological complications	13 (15.9)
Tumor	29 (35.4)
Skin tumors	8 (9.8)
Lymphomas	6 (7.3)
Solid tumors	15 (18.3)

CKD: chronic kidney disease, IS: immunosuppression.

**Table 3 jcm-12-06546-t003:** Information on fibrosis stages assessed by histological examination or Fibroscan, inflammation grade, and the degree of fat before, 1 year after IS disconsolation, and before IS reinstitution. F: fibrosis stage, I: inflammation grade.

			Before	After	*p*
Fibrosis stage	tolerant	0	9 (18.8)	6 (15.4)	1.000
	F-stage available: Before weaning; *n* = 48 After 1 IS-free year; *n* = 39	1	23 (47.9)	22 (56.4)	
		2	11 (22.9)	8 (20.5)	
		3	5 (10.4)	3 (7.7)	
		4	0 (0.0)	0 (0.0)	
	intolerant	0	5 (15.2)	5 (19.2)	0.206
	F-stage available: Before weaning; *n* = 33 Before IS reinstitution; *n* = 26	1	19 (57.6)	12 (46.2)	
		2	8 (24.2)	7 (26.9)	
		3	1 (3.0)	2 (7.7)	
		4	0 (0.0)	0 (0.0)	
	tolerant	0	11 (22.9)	7 (20.0)	0.593
Inflammation grade	I-grade available: Before weaning; *n* = 48 After 1 IS-free year; *n* = 35	1	22 (45.8)	22 (62.9)	
		2	14 (29.2)	5 (14.3)	
		3	1 (2.1)	1 (2.9)	
		4	0 (0.0)	0 (0.0)	
	intolerant	0	9 (27.3)	0 (0.0)	<0.001
	I-grade available Before weaning; *n* = 33 Before IS-reinstitution; *n* = 26	1	19 (57.6)	14 (53.8)	
		2	4 (12.1)	12 (46.2)	
		3	1 (3.0)	0 (0.0)	
		4	0 (0.0)	0 (0.0)	
Content of fat	tolerant	<5%	25 (52.1)	24 (68.6)	0.039
	Content of fat available Before weaning; *n* = 48 Before IS reinstitution; *n* = 35	5–33%	17 (35.4)	10 (28.6)	
		33–66%	2 (4.2)	1 (2.9)	
		>66%	4 (8.3)	0 (0.0)	
	intolerant	<5%	23 (69.7)	21 (80.8)	0.157
	Content of fat available Before weaning; *n* = 33 Before IS reinstitution; *n* = 26	5–33%	9 (27.3)	4 (15.4)	
		33–66%	1 (3.0)	1 (3.8)	
		>66%	0 (0.0)	0 (0.0)	

**Table 4 jcm-12-06546-t004:** Clinical parameters and variables analyzed for differences among tolerant and intolerant patients. Viral cirrhosis includes both HCV- (n = 36) and HBV- (n = 13) associated ESLD. D: donor, R: recipient.

Parameters	Units	Entire Cohort *n* = 82	Tolerant Patients *n* = 48	Intolerant Patients *n* = 34	*p*-Value
R-age at LT	Years (min–max)	52.4 (13–67)	53.2 (15.5–66.6)	51.6 (13.0–63.3)	0.221
D-age at LT	Years (min–max)	42.5 (12–89)	42.0 (12.0–71.0)	47.5 (12.0–89.0)	0.970
Time since LT	Years (min–max)	12.6 (0.4–30.7)	13.8 (5.5–30.7)	9.2 (0.4–29.0)	0.002
R-gender	female n (%)	33 (40.2)	12 (25.0)	21 (61.8)	0.001
	male n (%)	49 (59.8)	36 (75.0)	13 (38.2)	
D-gender	female n (%)	32 (39.0)	15 (31.3)	17 (50.0)	0.110
	male n (%)	50 (61.0)	33 (68.8)	17 (50.0)	
HCC	yes; n (%)	24 (29.3)	19 (39.6)	5 (17.4)	0.025
	no; n (%)	58 (70.7)	29 (60.4)	29 (85.3)	
Viral cirrhosis	yes; n (%)	49 (59.8)	32 (66.7)	17 (50.0)	0.171
	no; n (%)	33 (40.2)	16 (33.3)	17 (50.0)	
ACR	yes; n (%)	38 (46.3)	15 (31.3)	23 (67.6)	0.002
	no; n (%)	44 (53.7)	33 (68.8)	11 (32.4)	
Post transplant malignancy	yes; n (%)	29 (35.4)	23 (47.9)	6 (17.6)	0.005
	no; n (%)	53 (64.6)	25 (52.1)	28 (82.4)	
Malignancy (any)	yes; n (%)	40 (48.8)	31 (64.6)	9 (26.5)	<0.001
	no; n (%)	42 (51.2)	17 (35.4)	25 (73.5)	
Monotherapy	yes; n (%)	56 (68.3)	41 (85.4)	15 (44.1)	<0.001
	no; n (%)	26 (31.7)	7 (14.6)	19 (55.9)	
CNI mono	yes; n (%)	41 (50.0)	28 (58.3)	13 (38.2)	0.116
	no; n (%)	41 (50.0)	20 (41.7)	21 (61.8)	
CNI/MMF	yes; n (%)	24 (29.3)	6 (12.5)	18 (52.9)	<0.001
	no; n (%)	58 (70.7)	42 (87.5)	16 (47.1)	
MMF mono	yes; n (%)	13 (15.9)	12 (25.0)	1 (2.9)	0.012
	no; n (%)	69 (84.1)	36 (75.0)	33 (97.1)	

**Table 5 jcm-12-06546-t005:** Multivariable logistic regression analysis of factors contributing to tolerance in long-term liver transplantation. Wald statistics are shown for each predictor of tolerance as response variables of the penalized full regression model (d.f., effective degrees of freedom; OR, odds ratio; CI, confidence interval). Its linear predictor (LP) is calculated with the following estimated regression coefficients: LP = −2.9066 + 1.0024 × Time since LT + 0.4927 × post-LT malignancy + 1.1128 × HCC + 1.8440 × monotherapy + 1.1679 × recipient gender + 0.4215 × donor gender − 0.0144 × recipient age + 0.1197 × viral cirrhosis + 1.1664 × ACR. Adjusted odds ratios are shown and calculated from approximate coefficients of the simplified model: LP = −3.289 + 1.0176 × Time since LT + 1.1858 × HCC + 1.9716 × monotherapy + 1.3663 × recipient gender + 1.0731 × ACR.

Factor	χ^2^	*p*-Value	OR (95% CI)
Time since LT	2.65	0.103	2.77 (0.905–8.46)
post-LT malignancy	0.70	0.401	
HCC	2.89	0.089	3.27 (0.958–11.19)
monotherapy	9.11	0.003	7.18 (2.22–23.3)
recipient gender	4.11	0.043	3.92 (1.34–11.5)
donor gender	0.58	0.446	
recipient age	0.38	0.539	
viral cirrhosis	0.04	0.837	
ACR	4.67	0.031	2.92 (1.04–8.22)
TOTAL (d.f.)	26.35 (7.01)	0.002	

## Data Availability

The data presented in this study are available on request from the corresponding author. The data are not publicly available due to conditions of the ethics committee of our university.

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
