# Peer review of "How to Estimate the Probability of Tolerance Long-Term in Liver Transplant Recipients"

_jcm, 2023, doi:10.3390/jcm12206546_

Round 1

Reviewer 1 Report

Dear authors, you paper presents further evidence to support the safe withdrawal of IS from a subset of liver transplantation patients, especially with the contribution that a co-morbidity such as carcinoma does not preclude safe IS weaning and may therefore offer more effective cancer therapy in these patients.

There are some points that I want to make regarding the structure and clarity of the message in your paper, which could be improved.

Introduction.

Line 50

The importance of acute rejection for graft and patient survival is subject to temporal 

changes and is controversially discussed.  <- This statement does not add anything to the introduction and is arguably incorrect.

Immunologically neutral baseline diagnosis is a convoluted way of referencing LT patients with no signs of rejection.  

Line 53.

Sentence referencing paper number 7 incorrectly refers to  to which patient sub-population the predictive power of 79% represents, the paper suggests that sub-groups of stably transplanted patients (>3 years) can be identified by a combination of time since transplant, age and male sex. Of this further identified subgroup 79% predicted to achieve tolerance following weaning from IS.

The preceding description of the clinical acute response following surgery is not clearly expressed.

A general comment is that emotive imagery is overused in the introduction e.g. “avoidable iceberg” and “dramatic situation” as adjectives are not recommended.

Discussion.

Line 300 Second sentence too convoluted and over long (over 75 words).

Line 312 The statement that rejection is subordinate is not explained. What is rejection subordinate to?  Death without transplant, the management of another life threatening condition like cancer?

Line 316 Word error - nay instead of may.

Lines 336 - 349 and 350 - 355

These sentences discuss the hierarchy of risk to a patient who faces transplant rejection and another life threatening disease, such as malignancy.  The way these points are made are very hard to understand.  I suggest the points raised in the paragraph be reworked with the assistance of a native english speaker.

Lines 361 - 370 

Recapitulates points made above not adding much to the discussion.

Lines 376 - 378

The statement relating hepatic viral disease or tumours with tolerance due to a “blindness” of the immune system is just a supposition.

Do the authors have speculation of why increased age might be associated with a higher incidence of tolerance - like decreasing efficacy of adaptive immunity with age.

In general the Discussion could do with a reorganisation that focuses on the main points of the study.

  1. Re-emphasising the identification of a subset of patients with no signs of ACR that can weaned from IS with a high likelihood of success, predicted based the authors proposed 5 parameter score.
  2. Discussion of the balances of risks in patients that have multiple morbidities
  3. The fact that a pre-existing malignancy does not disadvantage IS weaning provides new opportunities to identify patients who can receive more effective treatment in the absence of IS.

The fact that of the >1500 patients identified, only 82 fitted the study inclusion criteria should be mentioned, that is this identification process is not broadly suitable for the majority of LT patients.

Included in comments above.

Author Response

Dear Reviewer,

thank you very much for your remarks and comments and the through revision. We have made every effort to address them and, where possible, incorporate them into the article. They are marked in yellow in the text.

Yours sincerely,

Dennis Eurich

__________________________________________________________________________________

Remark 1

Line 50

The importance of acute rejection for graft and patient survival is subject to temporal changes and is controversially discussed.  <- This statement does not add anything to the introduction and is arguably incorrect.

Reply: Thank you for the comment. Indeed, the sentence is somewhat nebulous and misleading. We have modified the sentence accordingly.

“The long-term impact of acute rejection on graft and patient survival is controversial.”      

__________________________________________________________________________________

Remark 2: Immunologically neutral baseline diagnosis is a convoluted way of referencing LT patients with no signs of rejection.  Sentence referencing paper number 7 incorrectly refers to  to which patient sub-population the predictive power of 79% represents, the paper suggests that sub-groups of stably transplanted patients (>3 years) can be identified by a combination of time since transplant, age and male sex. Of this further identified subgroup 79% predicted to achieve tolerance following weaning from IS.

Reply:  Thank you for the thoughtful comment. In the important work of Carlos Benitez (Ref. 7) et al, 98 patients were studied for tolerance, at least 90 of whom actually had "immunologically neutral" underlying disease (HBV, HCV, alcoholic cirrhosis, etc.).  It is an artificial collective term that has caused irritation. Therefore, the sentence was modified.

“In carefully selected patients without autoimmune-mediated underlying liver disease, the rate of spontaneous tolerance may be as high as 79% in the long-term.”

__________________________________________________________________________________

Remark 3: The preceding description of the clinical acute response following surgery is not clearly expressed.

Reply: We have made rewordings that we hope are clearer.

“While in the postoperative period the concern for rejection is dominant and a balance to infection is sought, the long-term transplant patient is in a stable state in which acute rejection is rare.”

__________________________________________________________________________________

Remark 4: A general comment is that emotive imagery is overused in the introduction e.g. “avoidable iceberg” and “dramatic situation” as adjectives are not recommended.

Reply: This is a valid point. We have deleted this sentence.

“(As the total volume of LT patients is increasing and heading towards an avoidable iceberg of long-term toxicity,)”

__________________________________________________________________________________

Remark 5: Line 300 Second sentence too convoluted and over long (over 75 words).

Reply: Thanks for the tips. The sentence is indeed too long, inhomogeneous and cluttered. We have made rewordings.

“The emergence of tolerance, defined as at least 12 months of IS freedom with clinical, laboratory, and histologic integrity of the graft, is a common phenomenon. In this retrospective analysis of real data from a large series of LT patients, a simple scoring system consisting of clinical parameters is presented to estimate the probability of tolerance.”

__________________________________________________________________________________

Remark 6: Line 312 The statement that rejection is subordinate is not explained. What is rejection subordinate to?  Death without transplant, the management of another life threatening condition like cancer?

Reply: Initial intent: “subordinate” in the overall context of comorbidities in the long-term course. We think, however, that at this point the importance of rejection should not be downplayed and deleted the word "subordinate".

__________________________________________________________________________________

Remark 7: Line 316 Word error - nay instead of may.

Reply Corrected

__________________________________________________________________________________

Remark 8: Lines 336 - 349 and 350 - 355

These sentences discuss the hierarchy of risk to a patient who faces transplant rejection and another life threatening disease, such as malignancy.  The way these points are made are very hard to understand.  I suggest the points raised in the paragraph be reworked with the assistance of a native english speaker. 

Reply: We appreciate the reviewers thorough revision. The following paragraphs were revised,a s suggested, with a fellow clinician and native English speaker.

“Clearly, the concern for graft survival must not supersede the survival of the recipient. Rather, integrated approach is important with regards to organ function and malignancy. Treatment options in case of malignancy are limited and overall survival is curtailed compared to a non-transplant setting as e.g. checkpoint inhibitors are contraindicated due to risk rejection by unmasking the antigen. Decreased immunosuppression improves patients’ prognosis and overall survival following HCC recurrence post LT. Reduction of IS to complete withdrawal would be highly beneficial in the clinical management of malignancies. Currently, there is scant data regarding type and extent of immunosuppressive therapy to prevent HCC recurrence or to improve long term prognosis.

The lack of integration of weaning into the principles of follow-up in clinical practice is clearly seen in oncological patients, although the negative influence of IS on the occurrence of post-transplant malignancies has been known for a long time.

This may have been as consequence of industry-sponsored IS trials, where minimizing IS was not considered and the principal focus was on reducing the rate of rejection. In the challenging circumstances of LT patient with de novo neoplasm, immunologic privilege of the liver might offer a solution without increased risk for the patient and graft survival. The data support this option, especially as maintaining the dose and type of immunosuppression is iatrogenic and results in reduced overall survival.”

__________________________________________________________________________________

Remark 9: Lines 361 - 370 

Recapitulates points made above not adding much to the discussion.

Reply: Lines were deleted.

__________________________________________________________________________________

Remark 10: Lines 376 – 378

The statement relating hepatic viral disease or tumours with tolerance due to a “blindness” of the immune system is just a supposition.

Reply: Thank you for pointing that out. You are correct that the statement is somewhat figurative and speculative and does not make a significant contribution. Therefore, the sentence was deleted.

__________________________________________________________________________________

Remark 11: Do the authors have speculation of why increased age might be associated with a higher incidence of tolerance - like decreasing efficacy of adaptive immunity with age.

Reply: That's a very interesting point, possibly making a conceptual transition from tolerance, which also occurs in the pediatric context as well as in elders, to immunosenescence, which actually corresponds to aging. Age-related changes in B cells are similar to changes in T cells. Age has an impact on the number of immune cells and on the repertoire of diversity of immunoglobulins and receptors, so that graft acceptance in the long-term is actually already facilitated by time and aging. These processes culminate in the futility of IS administration especially in these patients.      

__________________________________________________________________________________

Remark 12 In general the Discussion could do with a reorganisation that focuses on the main points of the study.

  1. Re-emphasising the identification of a subset of patients with no signs of ACR that can weaned from IS with a high likelihood of success, predicted based the authors proposed 5 parameter score.
  2. Discussion of the balances of risks in patients that have multiple morbidities
  3. The fact that a pre-existing malignancy does not disadvantage IS weaning provides new opportunities to identify patients who can receive more effective treatment in the absence of IS.

Reply: The discussion was modified and reorganized in many places. In doing so, we have made an effort to condense the content and have rewritten portions of the discussion. These sections have been highlighted in yellow for easier identification. 

__________________________________________________________________________________

Remark 13 The fact that of the >1500 patients identified, only 82 fitted the study inclusion criteria should be mentioned, that is this identification process is not broadly suitable for the majority of LT patients.

Reply: The 1500 patients mentioned represent the living collective, which in a way corresponds to the real world scenario with all the problems of the long-term course. Of course, the score cannot be applied to all 1500 patients, since a considerable proportion have been transplanted due to autoimmune-mediated underlying disease and have already dropped out. Nevertheless, the tolerance could be shown in different scenarios as proof that it is lived in reality. Of course, validation in larger populations is necessary. Further, weaning of IS has only become a clinical “desire” over the last decade and many physicians and patients as well do not “dare” to reduce IS to zero.

__________________________________________________________________________________

Reviewer 2 Report

is an interesting study that covers a significant problem in transplantation and therefore should be investigated in detail.

The first comment concerns postoperative care: the administration of immunosuppressive therapy (including the possible administration of drugs such as Basiliximab during surgery) is not clear 

The second comment is about age and indication for transplantation. The tolerance shown in a pediatric patient is definitely different from an adult recipient, so I believe the scores should be stratified by age. Same goes for indication.

The third comment is specifically about HCC. We know that not all hcc are the same, there are tumors that are much more aggressive than others in that sense variables like AFP and IRI should be taken into consideration. 

the last comment concerns the use of postoperative biopsies and/or fibroscans, which if I understand correctly were routinely performed at least 1 year after LT.  So in this case an invasive exam was used even on patients who did not show alterations on blood tests? on the other hand the sensitivity of an exam like fibroscan is likely to be a bit low after 1 year.  what is your opinion?

good

Author Response

Dear Reviewer,

We appreciate the constructive and thorough revision and in the following address all remarks in a point-by-point manner.

Yours sincerely,

Dennis Eurich

__________________________________________________________________________________

Remark 1: The first comment concerns postoperative care: the administration of immunosuppressive therapy (including the possible administration of drugs such as Basiliximab during surgery) is not clear

Reply: Our intra- and postoperative standard for the use of immunosuppression is as follows: intravenous application of methylprednsolone in the anhepatic phase, then starting with the first postoperative day application of tacrolimus 0. 1mg/kg divided into 2 doses with the target level between 8-15ng/ml depending on the underlying disease and steroid tapering regimen, in the course of months successive reduction of the tacrolimus dose to the range of about 5ng/ml with the completion of the first postoperative year, depending on the underlying disease and comorbidity addition of MMF (1-2g/d) or everolimus (target level 3-8ng/ml) in recommended dose. The use of induction by thymoglobulin or basiliximab is reserved for cases of retransplantation for immunological reasons and for patients with autoimmune-mediated underlying disease. In the case of oncological disease such as HCC, we do not use induction. We follow the principle: as much as necessary, as little as possible. In this way, many patient clusters are created with the individualized therapy up to complete IS discontinuation.

Since the report is not primarily concerned with examining the early phase, but rather the pattern of long-term progression with certain entry features, we have deliberately kept this block short.

__________________________________________________________________________________

Remark 2: The second comment is about age and indication for transplantation. The tolerance shown in a pediatric patient is definitely different from an adult recipient, so I believe the scores should be stratified by age. Same goes for indication.

Reply: This is an important point. In principle, you might be right that stratification should be done regarding age and indication groups. Due to the fact that the presented collective is not very large, I consider these steps impossible at least at this point. Furthermore, the tolerant patients do not differ significantly from the non-tolerant ones with respect to age in our cohort. This fact rather contrasts the other age-independent aspects. Similarly for the underlying disease: these were immunologically rather simple to the exclusion of autoimmune-mediated underlying diseases. The details are shown in Tables1 and 4.

This point coincides with Reviewer 1's comment: That's a very interesting point, possibly making a conceptual transition from tolerance, which also occurs in the pediatric context as well as in elders, to immunosenescence, which actually corresponds to aging. Age-related changes in B cells are similar to changes in T cells. Age has an impact on the number of immune cells and on the repertoire of diversity of immunoglobulins and receptors, so that graft acceptance in the long-term is actually already facilitated by time and aging. These processes culminate in the futility of IS administration especially in these patients.

__________________________________________________________________________________

Remark 3: The third comment is specifically about HCC. We know that not all hcc are the same, there are tumors that are much more aggressive than others in that sense variables like AFP and IRI should be taken into consideration.

Reply: Thanks for the comment. This cohort is patients who have been through the highest risk period for HCC recurrence. These patients underwent considerable selection already in terms of indication for TX and based on long-term outcome. We could not go into the tumor characteristics before TX in detail, although it would be interesting, but could show that the feature "HCC" is associated with tolerance and the development of other tumors in the long-term course.

__________________________________________________________________________________

Remark 4: the last comment concerns the use of postoperative biopsies and/or fibroscans, which if I understand correctly were routinely performed at least 1 year after LT.  So in this case an invasive exam was used even on patients who did not show alterations on blood tests? on the other hand the sensitivity of an exam like fibroscan is likely to be a bit low after 1 year.  what is your opinion?

Reply: Our center is one of the oldest in Germany and we still follow a rather strict assessment of graft health preferably by biopsy at rhythmic intervals for an indefinite period of time. Biopsy belongs to the gold standard of hepatic diagnostics, answers many questions and is able to show subclinical changes in still normal blood tests, which would lead to problems later on. Therefore, we refrain from biopsy only in rare cases and weigh the benefits and risks individually. As presented in the paper, histology is very important in defining tolerance.

Personally, I would never do without the first protocoll biopsy, which is performed after one year, because it often indicates the dynamics for the further course.

__________________________________________________________________________________

Round 2

Reviewer 2 Report

Exhaustive answers. I think this is a very important and topical subject. There remains the question of age stratification, which I think needs to be explored further with further studies, but this is definitely a good point to start.